# Two-Stage Fuzzy Traffic Congestion Detector

Gizem Erdinç *, Chiara Colombaroni and Gaetano Fusco

Department of Civil, Constructional and Environmental Engineering, Sapienza University of Rome, Via Eudossiana, 18, 00184 Rome, Italy
* Correspondence: gizem.erdinc@uniroma1.it

**Abstract:** This paper presents a two-stage fuzzy-logic application based on the Mamdani inference method to classify the observed road traffic conditions. It was tested using real data extracted from the Padua–Venice motorway in Italy, which contains a dense monitoring network that provides continuous measurements of flow, occupancy, and speed. The data collected indicate that the traffic flow characteristics of the road network are highly perturbed in oversaturated conditions, suggesting that a fuzzy approach might be more convenient than a deterministic one. Furthermore, since drivers have a vague notion of the traffic state, the fuzzy method seems more appropriate than the deterministic one for providing drivers with qualitative information about current traffic conditions. In the proposed method, the traffic states are analysed for each road section by relating them to average speed values modelled with fuzzy rules. An application using real data was carried out in Simulink MATLAB. The empirical results show that the proposed study performs well in estimation and classification.

**Keywords:** traffic state identification; fuzzy logic; congestion level; Mamdani inference

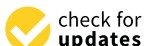



## 1. Introduction

Estimating traffic conditions has become one of the major problems in transportation engineering. If reliable information on traffic congestion is available, adequate and effective traffic control can be implemented to make traffic flow smoothly and improve the efficiency of surface transportation systems. Furthermore, drivers can be informed of states of congestion downstream, enabling them to change their route or simply drive more cautiously. Nevertheless, traffic congestion is a vague concept and there is still no universally accepted definition. It has either been defined as a state of traffic flow characterized by a high level of density and a low level of speed, as in [1], or it is defined on the basis of time, for example, as a function of delay [2], or, as in [3], as the additional time spent in the network when a vehicle is unable to drive at the free-flow speed level. On the other hand, congestion has been associated with insufficient road space, as an example of a situation occurring when the demand exceeds the supply on a road [4], or when vehicles obstruct each other because of an unbalanced speed–flow relation [5]. Finally, congestion has been defined in more than 10 different ways that are related to the demand-to-capacity ratio, delay and cost [6].

In addition to the definition of the concept of congestion, the thresholds of congestion states also vary between countries and societies. For example, according to the European DATEX II standards [7], congestion is defined at certain percentages of the road's free-flow speed level. In Asia, if the average level of speed drops to 19 mph for more than 2 h in a day and 10 days in a month, the Korea Highway Corporation accepts that congestion is occurring. This level is 25 mph in Japan [8]. Additionally, in Japan speed thresholds are also used to define the level of congestion [8]. According to the work of Skabardonis et al. [9] and Kwon et al. [10], congestion occurs when the speed is at a level of 60 mph on urban freeways, while the threshold is 45 mph in Minnesota [11]. In addition to these studies, Polus [12] determined congested conditions based on an occupancy value of more than 30%. Despite the existence of a huge literature on traffic congestion and great efforts to evaluate

it with the use of artificial intelligence methods [13], there is still no unified approach that is universally accepted [6]. This motivated us to continue studying this area and to propose our own approach.

The previous paragraphs show the amount of effort that has been devoted to the concept of congestion and its definition. Although there are technical specifications defining different congestion states for transport planning [14] and traffic information systems [7], several studies have demonstrated that drivers do not perceive congestion as a clear and precise notion [15]. Thus, qualitative or lexical information on traffic congestion, such as 'slow traffic' or 'queuing traffic', is rather vague for motorists, despite being able to be defined on the basis of a quantitative definition using precise intervals of a given traffic variable. Concerning the abovementioned examples, the Datex II European standard defines 'slow traffic' and 'queuing traffic' as conditions having an average speed between 25% and 75% and 10% and 25% of the free-flow level, respectively. This is a simple and rational method, but it is difficult for it to correspond to drivers' comprehension, because drivers do not perceive traffic conditions as being static and deterministic and do not have a unique and precise—quantitively defined—idea of them. Thus, the fuzzy approach, which is based on a non-univocal range definition of traffic conditions, seems to be a more appropriate approach to traffic congestion classification.

Furthermore, each traffic state has similar conditions at some level, resulting in there being some degree of similarity with each other traffic state, and each associated property has a level of uncertainty in real-world circumstances. On the other hand, to obtain reliable state information, a highly accurate prediction of short-term traffic parameters (i.e., occupancy, flow, speed) is a necessary step. In particular, the speed parameter is the most beneficial, since it is measured directly and is directly related to drivers' experiences. Current studies on short-term traffic speed prediction generally provide a prediction level, which is the average value of the parameter based on historical data [16]. However, there are many unpredictable factors that can influence the performance and accuracy of traffic speed level prediction. More importantly, the traffic stream corresponds to the multi-dimensional status of all parameters that are formed by driver behaviours. Therefore, an estimate is needed that can reflect the uncertainty and the noise effect on traffic parameters. Presently, the fuzzy approach makes it possible to consider the value ranges of the data series and to cultivate the integrity of the original data without the loss of any data information [17]. Thus, it has been argued that fuzzy qualitative definitions may better match drivers' perceptions [18].

This paper is organized as follows. The next section provides a review of the related works in the literature and highlights the original contribution of the method proposed. Section 3 illustrates the scope of the present work through a real example and defines the variables of the problem. Section 4 describes the proposed methodology for predicting and classifying traffic states. Section 5 illustrates the application of the method to an extensive set of traffic data collected on a stretch of motorway over an 8-month period. Section 6 provides a discussion of the results, while the conclusion is reported in Section 7.

## 2. Related Works

Fuzzy Logic (FL) is a qualitative approach based on approximation reasoning that is close to human thinking. A fuzzy system (FS) is a structure that represents inputs into the output universe of interest through fuzzy logic principles. In FSs, both subjective and objective inputs, which can be both numerical and/or linguistic data, are in consideration. It has been very popular for more than forty years in transport engineering applications such as speed control on expressways [19], signalization for traffic control [20,21], seaport [22] and transit [23] operations, lane-changing simulation models [24] and congestion-related applications [25–28]. In [25], the authors measured the level of congestion by using the same fuzzy approach with inputs such as speed reduction rate, the proportion of delay time within total travel time, and traffic volume to road capacity. Patel and Mukherjee [26] classified the traffic according to a fuzzified index of congestion and the average speed

level on the urban road network. Here, the congestion index was calculated as an output of the relationship between actual and free-flow travel time. The authors showed that the fuzzy approach was better at showing the real congested situation than other traditional congestion index values. Another fuzzy congestion evaluation study considering average speed as the input variable was presented by Hamad and Kikuchi [27]. They used travel speed, free-flow speed, and the proportion of very low speed in the total travel time as input variables to determine the congestion situation. Additionally, in [28], Kikuchi and Chakroborty studied a fuzzy approach for handling the uncertainty embedded in the definition of the level of service (LOS). They criticized the current HCM procedure, arguing that it does not accurately represent the notion of LOS as a user-perceived measure, and questioned whether a single measure (e.g., density) could capture all of the factors that affect LOS. Thus, they provided a framework that handles uncertainty under the different paradigms: deterministic, probabilistic, or possibilistic. Further studies in this field include [16,29–31]. On the basis of the described analyses, it is indicated that fuzzy-based applications have a preferable performance. However, recent experiments have generally been focused on detecting and testing abnormal events of traffic, and have rarely addressed the real-time estimation of the traffic state of the network.

The idea of detecting traffic congestion using the Mamdani-based fuzzy approach has already been studied and proven to be effective [32–34]. In [32], the authors used three inputs—the length of the lanes, the number of lanes, and flow data—to obtain the congestion level output. However, the experiment was only based on a one-week period of data; Kalinic and Keler [33] worked on a fuzzy method that compares two input sets: flow–density and occupancy–mean speed parameters for detecting traffic congestion; Kalinic and Krisp [34] presented a model containing only two inputs (flow and density), as in most classical traffic studies, which relate pairs of fundamental variables, with a few noticeable exceptions, like the application of Catastrophe theory to a 3D traffic state space, introduced in the 1980s [34], and which is still being studied and applied for the identification of traffic congestion [35,36].

With respect to the contributions of those reference works, we believe that traffic congestion should be evaluated in terms of speed, since it is a function of speed reduction in time. There are many studies considering speed as a parameter with or without others to determine the level of congestion on expressways and urban roads [8,25–27,29,31]. Specifically, in [8], the authors stated that a speed-based threshold has a greater impact on congestion than a threshold based on capacity does. From this point of view, we focused on the speed values. In this paper, we relate the classification of traffic states to the average speed variable instead of the density in order to obtain a perspective more compliant with the DATEX II European standard, which focuses on providing information to drivers, rather than being directed towards planning purposes. However, we also acknowledge that the extent of congestion is multi-dimensional, and the use of a single variable cannot give a decent assessment [37]. It is difficult to assume that there is a single value determining the entire traffic situation [38]. Furthermore, congestion is a state of traffic flow characterized by the fundamental variables together, which must be considered as a whole in the classification operation. Therefore, we developed a two-stage traffic congestion detector that is able to predict speed values by considering flow and density values, and which then provides a qualitative estimate of congestion that drivers will find more trustworthy, according to this reasoning.

With the aim of suggesting that a fuzzy approach might be more convenient than a deterministic one for catching the inherent uncertainty in the drivers' perception of traffic congestion when the flow characteristics are highly perturbed in oversaturated conditions, we tested our approach using data collected from the network on a motorway in Italy over a period of more than 8 months. The two-stage detector was able to predict the short-term speed values according to fuzzy rules and then classify the corresponding traffic states based on the EU DATEX II standard ranges.

### 3. Scope of the Work

#### 3.1. Observations of Traffic Variables

In this paper, we introduce a two-stage fuzzy method for traffic state identification and apply it to identify the traffic condition ranges on the Padua–Venice motorway network, which is composed of four branches of three-lane roads with separate carriageways, having a total length of about 74 km and including 16 intersections. Real traffic data were collected between 31 December 2018 and 30 August 2019, consisting of the following information: local unit code, code section counting, day type, road (section) ID, date, flow, density, and harmonic speed; these data were aggregated every 15 min. Figure 1 shows the monitoring sections of the relevant segments of the study network. The whole data set was examined statistically section by section; on the other hand, it is worth acknowledging that, due to temporary failures of some detectors, some of the data are missing. In this study, we studied the congestion for the time interval between 6:30 a.m. and 9 a.m., from Monday to Saturday, for 8 months. The application was run using MATLAB fuzzy logic toolbox R2020b and SIMULINK.

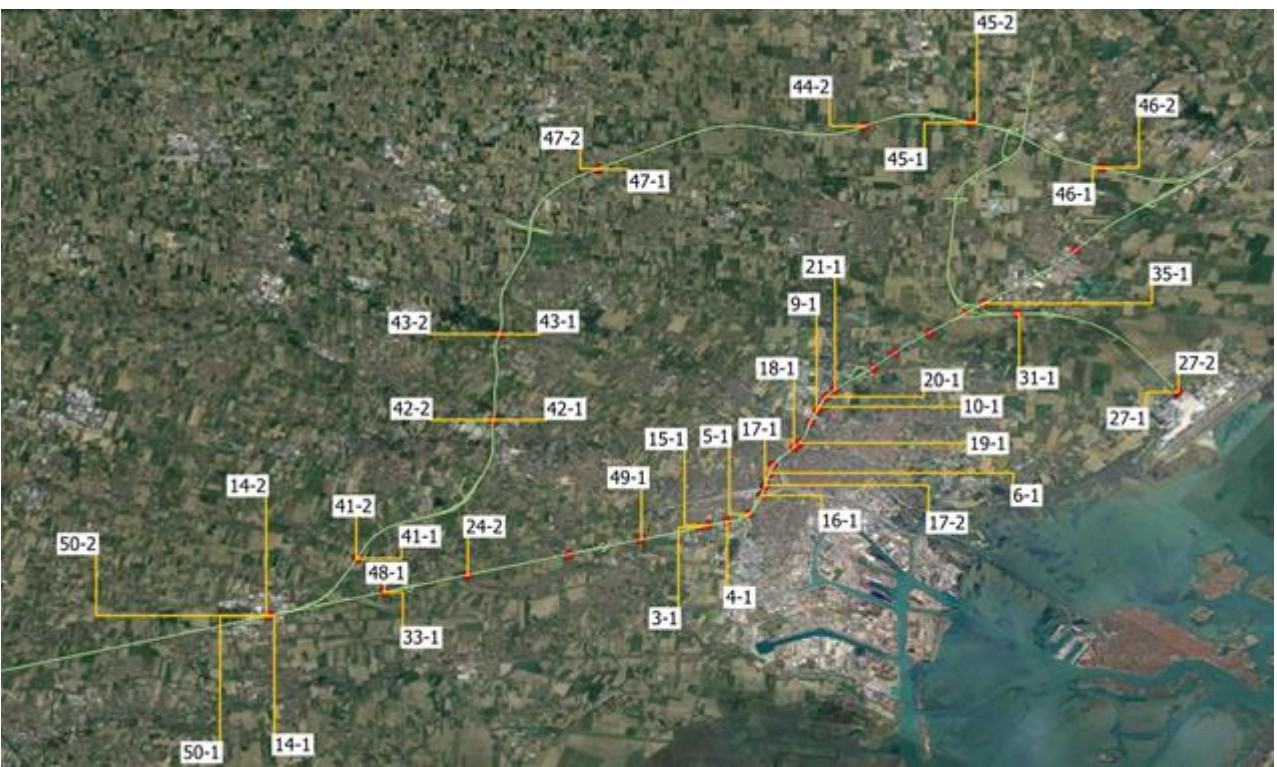

**Figure 1.** Identification codes of the sections.

Figure 2a–c illustrate the 15 min aggregated traffic observations collected at one of the road sections (section 41) over eight months. It is evident from the figures that the continuous collection of traffic data for such a long period introduces a huge noise component. The scope of the proposed model encompasses the simulation of the traffic states on the road and the derivation of the general relationship between the fundamental variables by applying the fuzzy Mamdani inference approach, regardless of their values [33]. Since different sections of road could have different traffic states, they have been modelled and simulated separately. The network contains two main branches, the North (Motorway) and the South (Tangential Highway). In this paper, we focus only on the Northern branch.

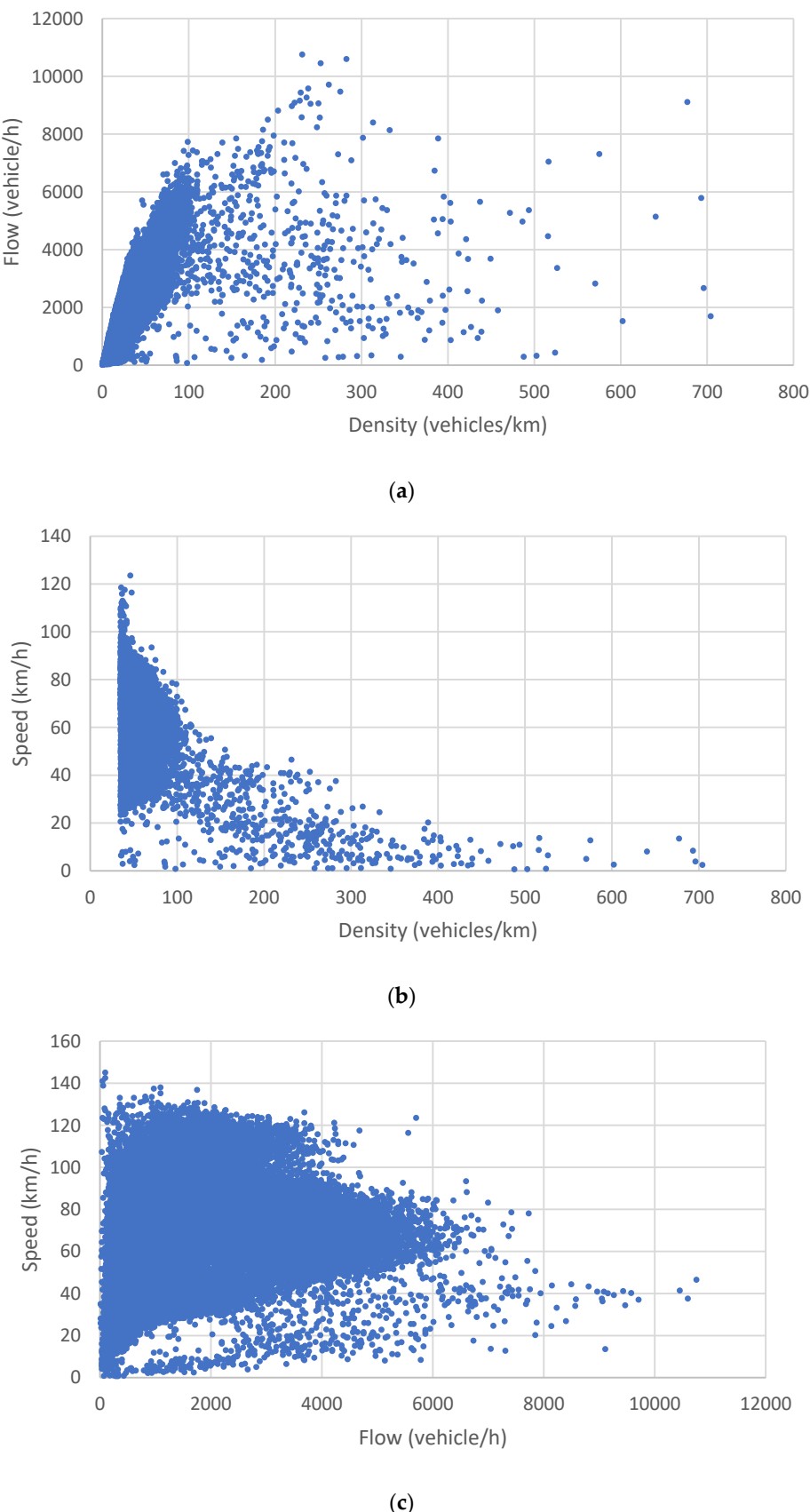

**Figure 2.** (**a**) Density–flow relationship; (**b**) density–speed relationship; (**c**) flow–speed relationship.

### 3.2. System Variables and Traffic State Classification

In traffic theory, three parameters are usually used to describe traffic flow characteristics: flow (f = vehicle/h), density (k = vehicles/km), and speed (v = km/h). The well-known relationship among the three parameters in a stationary state is expressed formally as Equation (1):

$$f = v \cdot k \tag{1}$$

Flow (f) refers to the number of vehicles passing a certain cross-section per time unit in a given time interval. Density (k) is the parameter most relevant to the congestion phenomenon, because it characterizes the quality of traffic operations, and it is more directly related to the manoeuvrability of vehicles in the traffic stream. It represents the number of cars on a unitary long road segment at a given instant in time. Speed (v) is another common congestion indicator, as it shows the degree of vehicle mobility on the road. It is worth remembering that congestion is a function of changes in speed. In this study, density is derived from the occupancy of the loop detectors, and flow and speed values are taken from the sensors.

While a traffic state is defined as a given tuple of values of fundamental variables, the definition of road congestion is ambiguous and not easy to define. A huge mass of traffic observations exist. Nevertheless, an example is presented here, the only aim of which is to provide an illustrative explanation of the concepts underlying the fuzzy approach introduced in this paper. As is well known, in the flow–density plane, direct traffic measurements illustrate two different patterns (Figure 2a): where the flow is almost linear with density, this represents limited deviations from the average speed in the low-density range; in the high-density range, a very noisy and sparse pattern occurs with an average decreasing trend of the flow with increasing density. It is well known that the high-density regime is characterized by unstable flow conditions, as determined by the microscopic mechanism underlying the traffic flow, where even slightly irregular driving manoeuvres lead to a stop-and-go regime. The observation of speed–density measures highlights in a clear way the decreasing trend of speed with density (Figure 2b) and explains the decrease in flow in the high-density regime by virtue of Equation (1). While the speed–density relationship is suitable for model calibration because of its monotone trend, the fundamental diagram can be more conveniently used to determine traffic states and classify their congestion levels.

In general, by conceiving of the flow–density relationship as a bell-shaped curve, traffic conditions can be divided into different states that describe the change from free-flow to congested conditions, with particular attention being paid to the critical point at which the curve's trend becomes inverted, and the flow reaches its capacity. Many studies have faced this problem using different approaches and for different purposes. While applications devoted to planning, like the US Highway Capacity Manual [14], and management, like the EU DATEX [7], consider five traffic states corresponding to many levels of services, incident detection algorithms focus on the simplest distinction between congested flow and non-congested flow. Theoretical studies have considered different numbers of traffic classes, ranging from two [39], through three [40], and four [41], to five [42]. In our view, evaluating the states according to the EU DATEX standard, we divided the range of traffic conditions into five features. Table 1 summarizes qualitative parameter definitions with respect to the traffic fundamental diagram.

The first state is the smooth state, in which condition the average speed is very high (more than 90% of the free-flow level), and the flow and density are very low. In this condition, drivers are hardly influenced by the vehicles ahead of and behind them, and they are able to freely drive the way they want. The second state is the intense state. In this state, the average speed is still high (between 75 and 90% of the free-flow level), and flow is low, but density is medium. When the density increases further, and the flow increases to the traffic capacity, the state would be classified as the slow state. Under this condition, the road can be fully used, and drivers are still able to drive with high variability of speed (average speed between 25 and 75% of the free-flow level), but with a reduced level of

freedom. The fourth state is the queuing state. This state is characterized by low values of speed (between 10 and 25% of the free-flow level), and high values of density and flow; this may lead to unstable traffic conditions. The fifth state is the stationary state. The density is very high, and speed is very low (average speed < 10% of the free-flow level). It is characterized by stop-and-go conditions, which cause traffic jams. Table 1 summarizes the qualitative parameter definitions related to the fundamental traffic diagram.

**Table 1.** Traffic state classification summary.

| State | f | k | v |
|---|---|---|---|
| Smooth | Very low | Very low | Very High |
| Intense | Low | Medium | High |
| Slow | Medium | Medium-High | Low |
| Queuing | High | High | Low |
| Stationary | Low | Very high | Very low |

## 4. Methodology: Two-Stage Fuzzy Traffic Congestion Detector

In this paper, we present a two-stage fuzzy-based approach integrating a short-term predictor (first step) and a classifier (second step). The general framework of the method is given in Figure 3. We believe that a method that assesses speed values can improve the effectiveness of detection because it is better able to characterize unstable conditions when fundamental variables are affected by rapid changes that violate the relationships that hold under stationary conditions. However, in order to do so, we acknowledge that the method used must be able to successfully reflect the relationships between the fundamental variables. Because these fundamental variables are essential for traffic engineers in several stages of a project, such as planning, design and implementation [43]. Therefore, we wanted to investigate the performance of the Mamdani-based fuzzy logic approach at modelling the relationships between fundamental variables. The best way to test this is by estimating the value of the third variable using the other two. Following this idea, we developed a method—the two-stage traffic congestion detector—that takes as input the real traffic data for each road section and identifies the relationships between the fundamental variables (f-k-v), before estimating the speed values and determining the level of congestion according to the speeds. We focused on the speed variable instead of density in general, because speed can be measured directly, and is directly related to drivers' experiences and to the total time spent on the network, which is a frequently used performance indicator for road traffic [8,24–26,28,30].

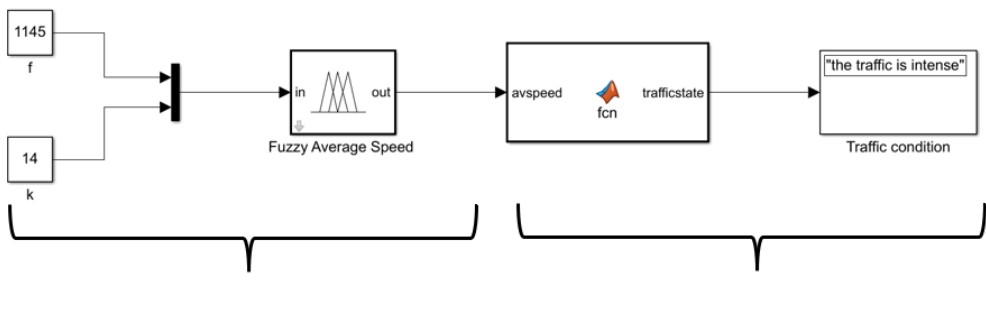

**The fuzzy predictor (First step)**          **The classifier (Second step)**

**Figure 3.** General framework of the method.

In the first step, fuzzy average speed values are computed using both flow and density inputs. After calculating the average speed values, in the second step, the traffic conditions related to them are classified. The whole model is simulated in Simulink MATLAB. In a

different view from [32,34,37], the average speed is applied to detect the traffic states. This is the output parameter of the first part and the input parameter of the second part. Moreover, unlike the above-mentioned papers [32,34,37], the fuzzy model is applied here to predict traffic conditions in the next time interval. Indeed, it has been demonstrated that traffic prediction models can be improved when matching them with a preliminary classification of the traffic state [44]. The explicit speed prediction during the first step makes it possible to assess the quality of the predictions according to standard statistical methods; the final qualitative output supplies simple information that can be communicated to drivers and traffic operators. Additionally, it is expected that giving estimations resulting in a logical process between variables could enhance the quality of congestion assessment, since many works [25–27] have used compound measured values as inputs to determine the level of congestion.

### 4.1. First Step: Short-Term Average Speed Prediction

The first step of the general method was built in MATLAB and run using SIMULINK, as shown in Figure 4. The fuzzy model named 'Fuzzy Average Speed', presented in the middle of the figure, calculates the average speed values for the next time interval by using flow and density information.

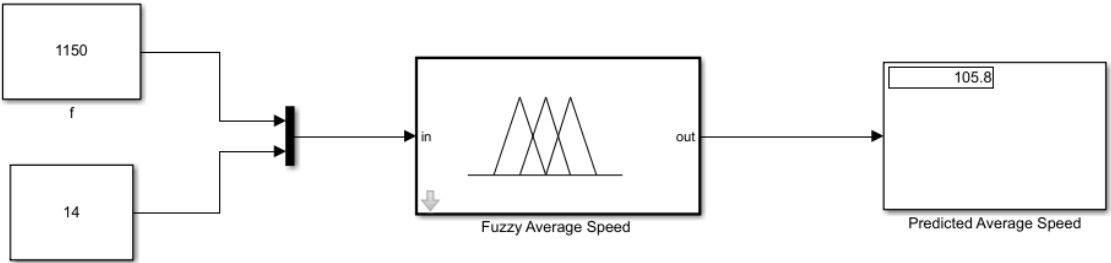

**Figure 4.** First step: the fuzzy predictor.

In the 'Fuzzy Average Speed' model, we define two input parameters (flow and density) and one output parameter (average speed), as shown in Figure 5. We set $n_{xi}$ (i: variable) linguistic terms for each of them.

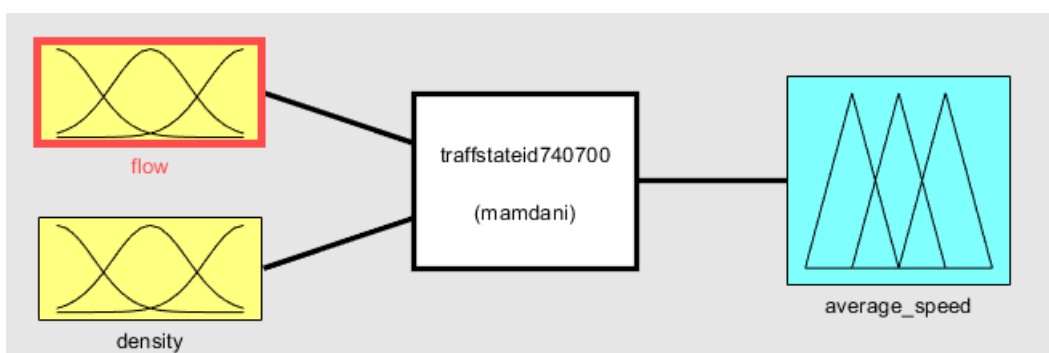

**Figure 5.** The fuzzy average speed model.

To represent the wide range of data, we set $n_{x1} = n_{x2} = n_{x3} = 5$ linguistic terms as: F = $\{FF, RFF, AF, CF, VCF\}$ and K = $\{VLD, LD, MD, HD, VHD\}$
The output variable is also classified as: V = $\{VS, S, A, F, VF\}$
Explanation of variables:

FF: Free Flow, RFF: Reasonably Free Flow, AF: Average Flow, CF: Congested Flow and VCF: Very Congested Flow
VLD: Very Low Density, LD: Low Density, MD: Medium Density, HD: High Density and VHD: Very High Density

VS: Very Slow, S: Slow, A: Average, F: Fast and VF: Very Fast

After defining the input and output, the 'fuzzification' stage should be carried out. During this stage, all variables are fuzzified by transferring the crisp numerical values into membership degrees of the fuzzy set. This is done using membership functions, which give the quantity of the degree of belongingness of a numerical value to the related fuzzy set in a closed interval [0;1]. Here, 1 expresses full membership, and 0 denotes non-membership. Among the many possible functions, the most common ones are triangular [32], trapezoidal [33,34,37], Gaussian, generalized bell, and sigmoidal. In this paper, triangular membership functions are used (Figure 6a–c), since they are one of the most widely used examples, and they can effectively reflect the characteristics of the fuzzy sets used here.

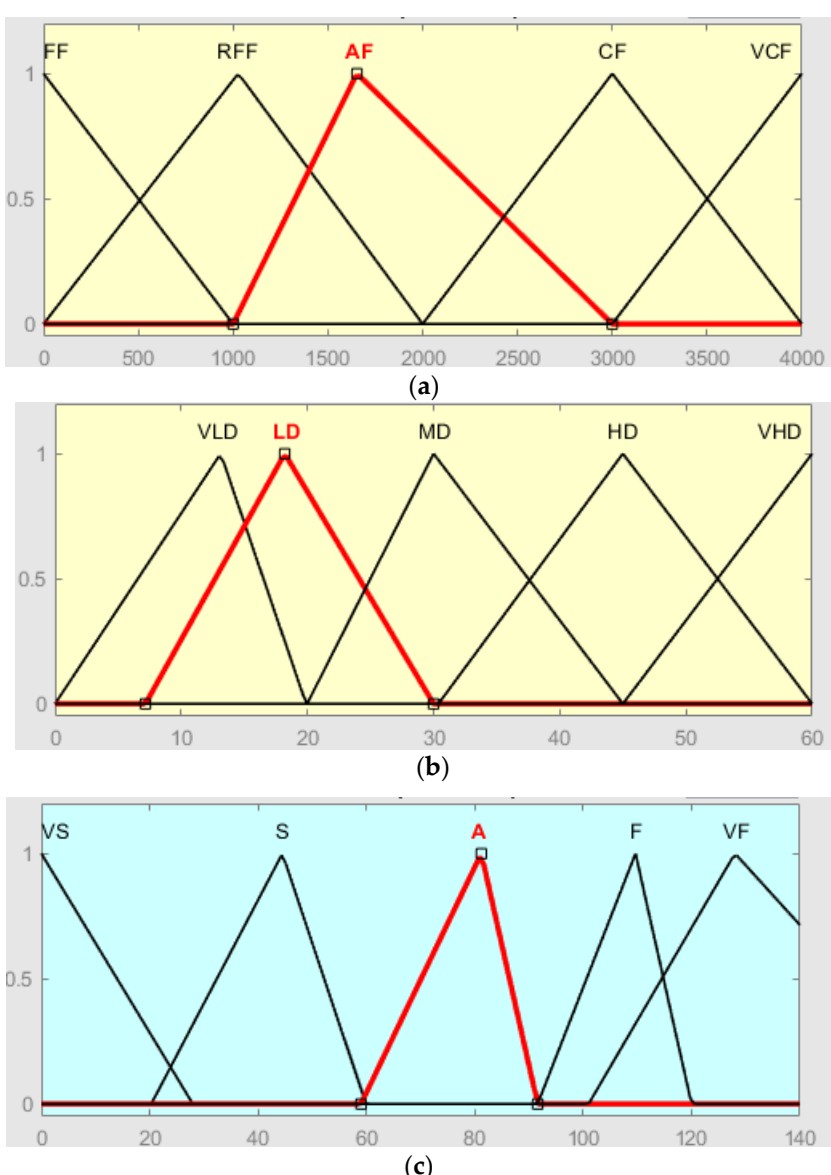

**Figure 6.** (**a**) Membership function μ of flow in the range [0;4000]. (**b**) Membership function μ of density in the range [0;60]. (**c**) Membership function μ of speed in the range [0;180].

The boundary sets of the membership functions are relevant in this regard, since they can affect the degree of belongingness of the value to the fuzzy set. Here, they are set mostly with reference to Table 1 in [37]. For example, the density variable is classified as 'Low Density' in the range between 7 and 30 vehicles/km, as in [37], with the subjective

addition of the middle value being equal to 18 vehicles/km, which is associated with LOS B, C, D.

As an example of the mathematical form, the membership function of the *Low-density* variable is provided in Equation (2):

$$\mu LD(k) = \begin{cases} \frac{k-7}{18-7}, 7 \le k < 18 \\ \frac{30-k}{30-18}, 18 \le k < 30 \\ 0, k < 7 \text{ or } k \ge 30 \end{cases} \tag{2}$$

For the output variable fuzzy sets, ref [7] was examined.

As the core of the method, the input–output relationship should be modelled to build the inference and a nonlinear surface model with specific rules, which demonstrates how input variables are reflected in the output universe. In the literature, two types of fuzzy inference system have been described: Mamdani and Sugeno types. The prime difference between them is how the outputs are determined. Currently, Mamdani's fuzzy inference method is popular for use in complex problems, while Sugeno-type systems are effective only in cases that have either linear or constant output membership functions. In this study, Mamdani-based inference is used. The rules for section ID 740700 are given below, and they can be stated mathematically as:

$$(F \bullet F(f)) \Theta min(K \bullet K(k)) \rightarrow (A \bullet A(f, k)), \tag{3}$$

where the symbol $\bullet$ states the linguistic term IS, the symbol $\Theta min$ states the logic operator AND.

1. If density is 'Very Low' then average speed is 'Average'.
2. If density is 'Very Low' then average speed is 'Fast'.
3. If density is 'Very Low' then average speed is 'Very Fast'.
4. If density is 'High' then average speed is 'Slow'.
5. If density is 'Very High' then average speed is 'Very Slow'.
6. If flow is 'Free' then average speed is 'Average'.
7. If flow is 'Free' then average speed is 'Fast'.
8. If flow is 'Free' then average speed is 'Very Fast'.
9. If flow is 'Reasonably Free' then average speed is 'Average'.
10. If flow is 'Reasonably Free' then average speed is 'Fast'.
11. If flow is 'Congested' then average speed is 'Fast'.
12. If flow is 'Congested' then average speed is 'Average'.
13. If flow is 'Congested' then average speed is 'Slow'.
14. If flow is 'Very Congested' then average speed is 'Average'.
15. If density is 'Very High' then average speed is 'Slow'.
16. If density is 'Very High' then average speed is 'Average'.
17. If density is 'High' then average speed is 'Very Slow'.
18. If density is 'High' then average speed is 'Average'.
19. If flow is 'Average' then average speed is 'Very Fast'.
20. If flow is 'Average' then average speed is 'Fast'.
21. If flow is 'Reasonably Free' and density is 'Very Low' then average speed is 'Very Fast'.
22. If flow is 'Reasonably Free' and density is 'Low' then average speed is 'Fast'.
23. If flow is 'Average' and density is 'Low' then average speed is 'Fast'.
24. If density is 'Average' and density is 'Very Low' then average speed is 'Very Fast'.

When the inputs are defined with more than one fuzzy set, as in rules 21 to 24, the membership values must be associated with obtaining a unique result. This is done by including the AND operator between linguistic information. Here, the operator provides a

minimum condition that must be met in the conditional IF statement to be fulfilled. It can be formalized as:

$$\mu_{F \cap K} = \text{MIN} (\mu(F), \mu(K)) \tag{4}$$

At this point, each IF–THEN rule refers to a fuzzy set with the corresponding belonging membership values, which must be accumulated into a single fuzzy set. The MAX operator is one of the most widely used operators for this process. After this aggregation operation, the fuzzy set must be de-fuzzified. In this study, we used the centroid method, since it is the most widely applied.

### 4.2. The Second Step: Classification

For the second step, traffic congestion states were identified using the second fuzzy model based on the speed value provided as output by the first fuzzy step (Figure 7). The same traffic classes were defined by assigning the rules below according to the DATEX II standard [7] to express the states:

If average speed < 10% of free-flow level, then the traffic is stationary;
If 10% ≤ average speed < 25% of free-flow level, then the traffic is queuing;
If 25% ≤ average speed < 75% of free-flow level, then the traffic is slow;
If 75% ≤ average speed < 90% of free-flow level, then the traffic is intense;
If average speed ≥ 90% of free-flow level, then the traffic is smooth.
In this study, the free-flow speed level is 140 km/h.

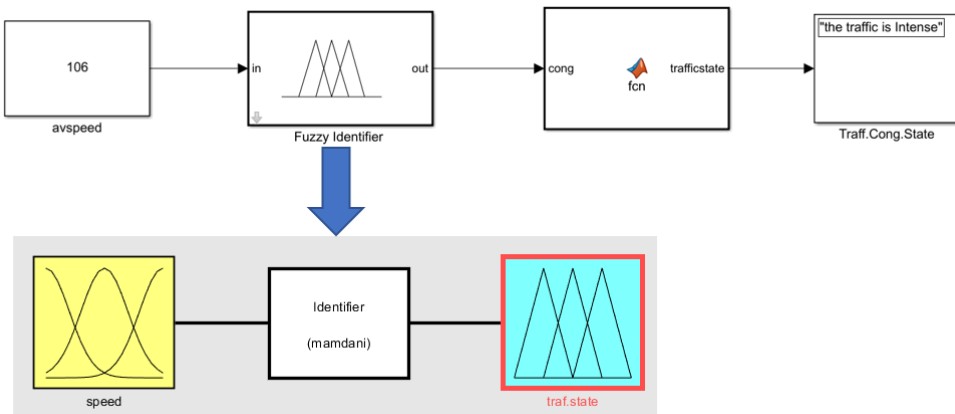

**Figure 7.** Second step: the classifier.

The membership functions of the speed variables are the same as in the first step. The universe discourse of the traffic congestion state variable is normalized to the scale [0;1], and the membership functions of the states are given in Figure 8.

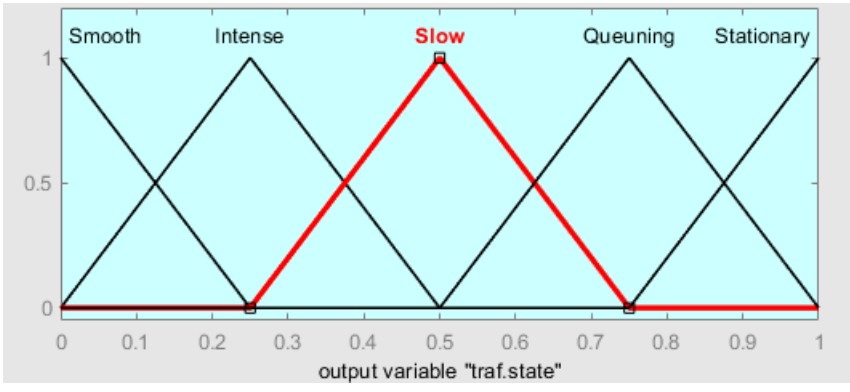

**Figure 8.** Membership function µ of traffic congestion state in the range [0;1].

Finally, the two steps were combined, and the whole system was created as shown in Figure 3. This system estimates the traffic congestion state on the basis of the speed values that have been predicted by the flow and density information that has been fed, while considering the qualitative description of traffic.

## 5. Application Results

The first step of the application consisted of computing speed predictions for 15 min time intervals between 6:30 and 9:00 for the six workdays of every week from Monday to Saturday for 8 months for all sections along the highway. The reason for this computation was to test the performance of the speed predictor and compare its results to the observed values. Table 2 shows average flow, average density, observed average speed, and fuzzy predicted average speed values, respectively, for section ID 740700 as an example. As mentioned previously, the average speed that a traffic flow can have for 15 min is related to the cumulative flow and density values on the road. Hence, the predictor considers the average values of flow and density belonging to the previous 15 min time interval (denoted as the input timeline in Table 2), and then predicts the average speed value for the next 15 min interval (output timeline). For example, when predicting the average speed for the time interval 6:30–6:45, the average flow and average density information of the time interval 6:15–6:30 is used.

**Table 2.** Fuzzy average speed predictions for section ID 740700 with the subset of data for time intervals between 6:30 a.m. and 9:00 a.m.

| No | Input Timeline | Flow | Density | Observed Speed | Output Timeline | Fuzzy Predicted Speed |
|---|---|---|---|---|---|---|
| 1 | 6:15–6:30 | 1150.94 | 13.94 | 105.7 | 6:30–6:45 | 105.8 |
| 2 | 6:30–6:45 | 1150.12 | 13.92 | 105.8 | 6:45–7:00 | 105.8 |
| 3 | 6:45–7:00 | 1148.94 | 13.89 | 105.9 | 7:00–7:15 | 105.9 |
| 4 | 7:00–7:15 | 1147.52 | 13.85 | 106.0 | 7:15–7:30 | 106 |
| 5 | 7:15–7:30 | 1145.85 | 13.83 | 106.0 | 7:30–7:45 | 106 |
| 6 | 7:30–7:45 | 1144.02 | 13.80 | 106.1 | 7:45–8:00 | 106 |
| 7 | 7:45–8:00 | 1141.99 | 13.77 | 106.1 | 8:00–8:15 | 106 |
| 8 | 8:00–8:15 | 1139.65 | 13.73 | 106.2 | 8:15–8:30 | 106 |
| 9 | 8:15–8:30 | 1137.32 | 13.70 | 106.2 | 8:30–8:45 | 106.1 |
| 10 | 8:30–8:45 | 1135.07 | 13.67 | 106.3 | 8:45–9:00 | 106.1 |
| | average | (~1145) | (~14) | 106 | | 105.97 (~106) |

The application of the prediction method was repeated for all sections. The corresponding predicted values were computed and compared with the observed ones. The differences between these values are reported in Figure 9, which also illustrates the standard deviation of the measures, making it possible to compare the prediction errors to the variability extents of the variables. Along the highway, the observed average speed values ranged between 93 and 108 km/h, while the predictions ranged from 92 to 107 km/h. In all sections, the errors were within the range of the standard deviations. The prediction results were very close to the observed speed values. It is important to specify that the focus of this study was on classifying traffic using a qualitative method, rather than on the application of a prediction method. The predicted short-term rates are the intermediate outputs of this study, and they are given here only to highlight that a qualitative approach can be used effectively in unclear and vague environments.

After obtaining the predicted speed values, the traffic states were identified section by section. With reference to road section ID 740700 as an example, when flow f = 1145 vehicles/hour and density k = 14 vehicles/km, the predicted speed was 106 km/h. Under these circumstances, traffic can be considered to be in an *intense* situation. Table 3 and Figure 10 show the traffic conditions for all sections of the highway.

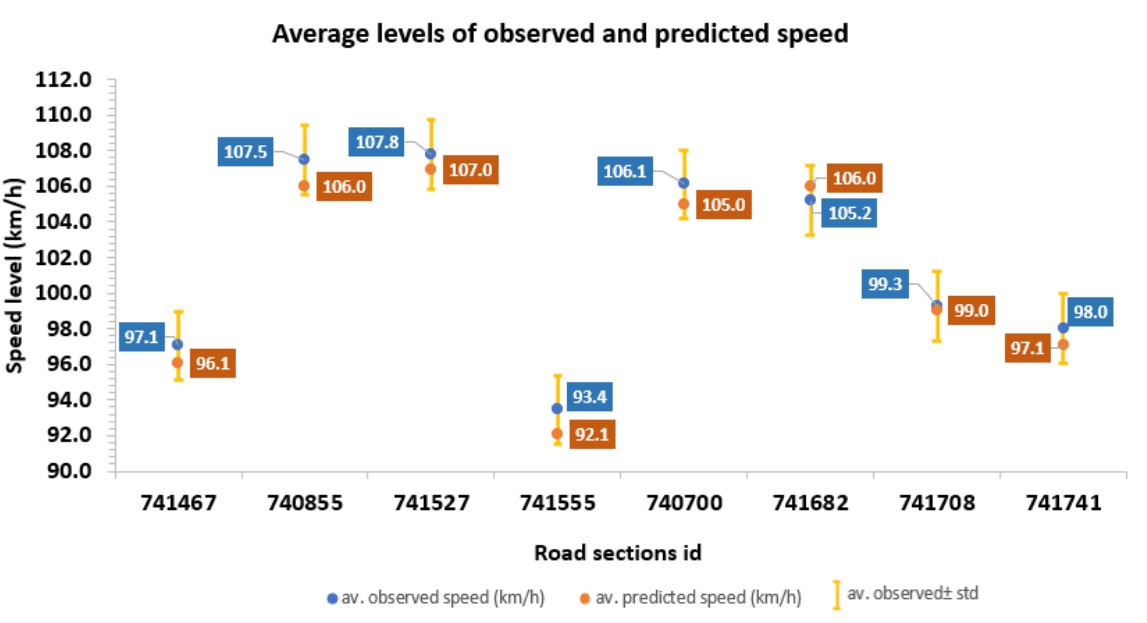

**Figure 9.** Average speeds for all sections.

**Table 3.** Identified traffic states for each section with the subset of data with time intervals between 6:30 a.m. and 9:00 a.m.

| Section Number | Road Section | Average of the Observed Intervals | | | | Traffic State |
|---|---|---|---|---|---|---|
| ID | ID | Flow (veh/Hour/Lane) | Density (veh/km/Lane) | Observed Speed (km/h) | Predicted Speed (km/h) | |
| 41-1 | 741467 | 1531 | 17 | 97 | 96 | Slow |
| 42-2 | 740855 | 1480 | 15 | 107 | 106 | Intense |
| 42-1 | 741527 | 1475 | 15 | 108 | 107 | Intense |
| 43-1 | 741555 | 1453 | 17 | 93 | 92 | Slow |
| 44-2 | 740700 | 1145 | 14 | 106 | 106 | Intense |
| 44-1 | 741682 | 1042 | 13 | 105 | 102 | Intense |
| 45-1 | 741708 | 1432 | 15 | 99 | 99 | Intense |
| 46-1 | 741741 | 973 | 11 | 98 | 97 | Intense |

The observations of traffic states along the entire highway show a similar pattern, with *intense* or *slow* state levels. There is a smooth transition to a *slow* traffic state in section 43 and an opposite one to an *intense* state in sections 44 and 42. The threshold for passing into the *slow* state from the *intense* condition is set at flows higher than 1400 vehicles/h and densities higher than 17 vehicles/km. In this case, the average speed decreases by around 15% compared to its previous level. This means that drivers are able to drive at around 67% of the free-flow speed level of the road. In such a traffic state, flow and density rates decrease to around 1150 vehicles/h and 14 vehicles/km, respectively. This allows the speed level to increase to around 106.0 km/h, which is more than 75% of the free-flow level, giving drivers more freedom to manoeuvre.

There is no severe congestion or occurrence of a *stationary* state along the highway, because an average flow with a low density allows fast speed with high membership and average–very fast speed with low memberships, a situation that generally occurs in *intense* states.

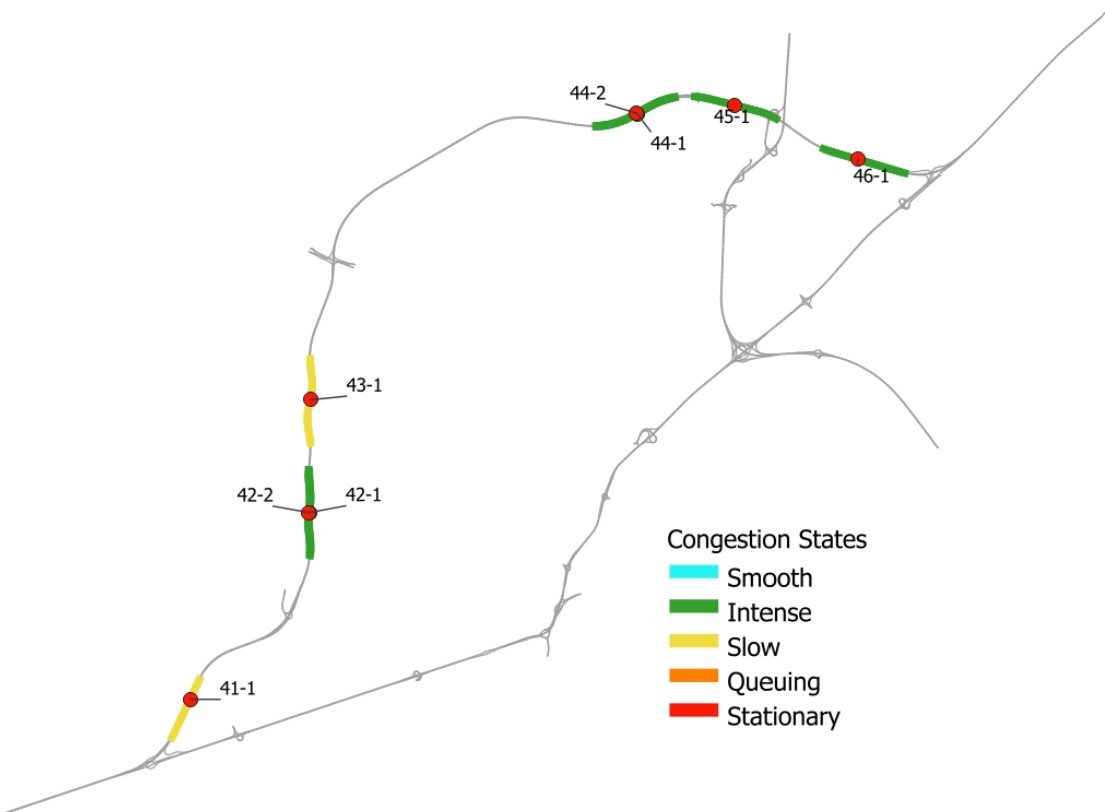

**Figure 10.** The situation of the highway with the subset of data with time intervals between 6:30 a.m. and 9:00 a.m.

## 6. Discussion

### 6.1. Comparison with the Levels of Service Definitions of the US Highway Capacity Manual

The contribution of our method is further discussed on the basis of a comparison of our results with the corresponding estimates of LOSs obtained according to the HCM method [13], where the levels of service traffic are defined based on the diagram presented in Figure 11. Since the HCM speed–flow curves are derived from observations conducted in the USA, they should be applied to an Italian motorway with great caution, and should be considered only as a comparison between methodologies. In fact, the motorway under study has a speed limit of 130 km/h, which is a value that exceeds the ranges considered by the HCM. However, if an extrapolation of the trend sketched by the HCM curves is considered admissible, then an LOS can be attributed to the observed traffic states. However, this exercise reveals that the theoretical model based on a univariate relation between speed and flow and the state Equation (1) is inconsistent with the joint observations of the three state variables reported in Table 2. In fact, if we consider density as being the indicator for LOS identification, the observed traffic states, which range from 11 veh/km/lane (that is, 18 veh/mi/lane) to 17 veh/km/lane (that is, 27 veh/mi/lane), would be assigned to LOS C or LOS D. However, if we consider the values of flow, which range from 973 veh/h/lane to 1531 veh/h/lane, the corresponding LOS will vary from LOS B to LOS C. The fuzzy method classified these traffic conditions as either *intense* or *slow*. Since the fuzzy method considers the three state variables together, it encompasses the unavoidable uncertainty in traffic state identification and eliminates classification ambiguity due to the inherent errors introduced by the steady-state assumptions underlying the HCM methodology. Additionally, the qualitative definition is more intuitive than a scale classification, since it uses common adjectives, making it more suitable for providing information to drivers.

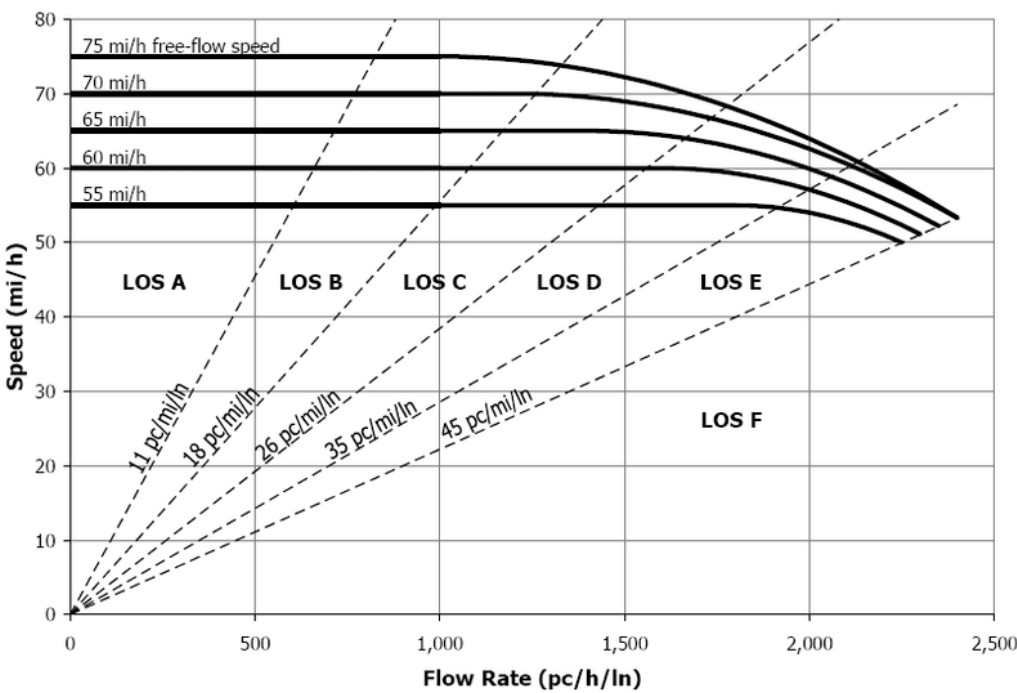

**Figure 11.** Speed–flow–density relationship [14].

It is worth remembering that, in this study, we claim that fuzzy logic can be used even in ambiguous environments. This is an added value of the two-stage method, because it allows the speed value to be predicted, but then provides a qualitative estimate of congestion, which will be seen as more trustworthy by the drivers, according to this reasoning. Hence, we believe that in the case of highly noisy and stochastic environments, a qualitative fuzzy estimate approach would be more suitable and wiser than a point estimate.

### 6.2. Limitations

While we believe that the proposed study provides effective approaches to traffic management due to advantages such as ease of application, being close to human thinking, and flexible design in uncertain and subjective cases, the two-stage fuzzy traffic congestion detector has certain limitations, as follows:

- The current results are limited to showing severe or temporal congestion corresponding to the static situation of the case study, and lack of information on relevant aspects that can affect the level of congestion, such as accidents, road quality, maintenance works, etc.
- The whole system is built on a fuzzy logic approach that can show the best performance on the basis of well-defined rules, proper membership functions, and clear input–output relations. However, each process requires a long learning period, as well as experience in the field.
- The two-stage fuzzy traffic congestion detector has a non-linear and complex behaviour.
- In this study, we worked on data collected every 15 min. This timeframe can be criticized in terms of its effectiveness for characterizing the trend of rate changes. This is a fair criticism. In future studies, the present method will be tested using shorter time intervals, provided that reliable data are available.

### 7. Conclusions and Remarks for Future Works

In this paper, we addressed the traffic state identification problem and proposed a fuzzy logic-based method that was then applied to a much larger real dataset. We related

the states' output to the average speed values, which constituted a pre-output of a fuzzy model. Because of the necessity of performing a multi-dimensional assessment, and because we had all three traffic-flow variables, we evaluated the ability of fuzzy logic to model the variables and to perform driving behaviour modelling. To test this, we presented a two-stage traffic congestion detector, which was capable of modelling different traffic states. With a proper rule-based design, the two-stage traffic congestion detector can be employed in practice as a support tool for formulating control actions on expressways under boisterous conditions. This level of support could lead the way in studying traffic breakdown-related alerts and intelligent early warning systems, with potential benefits in dealing with congestion-related traffic problems. In contrast to traditional methods used for the detection of traffic congestion, the two-stage detector enables the prediction of the speed value, and then provides a qualitative estimate of congestion, which drivers find more trustworthy. Therefore, a different perspective was given for the motorway traffic control and traffic induction literature that is based on the speed variable, and it was shown that the fuzzy approach can be used for short-term prediction as well. However, this study was limited to showing severe and/or temporal congestion, because of the static situation of the case study. In future works, the model will be extended in consideration of its limited static situation. Additionally, to make it more convenient for designing control actions on motorways, unexpected situations such as accidents, weather changes, etc., will be addressed.

**Author Contributions:** Conceptualization, G.E. and G.F.; methodology, G.E.; software, G.E.; validation, G.E. and G.F.; formal analysis, G.E.; investigation, G.E.; resources, C.C.; data curation, G.E.; writing—original draft preparation, G.E.; writing—review and editing, G.E. and G.F.; visualization, G.E.; supervision, G.F. and C.C.; project administration, G.F.; funding acquisition, G.F. All authors have read and agreed to the published version of the manuscript.

**Funding:** This research received no external funding.

**Institutional Review Board Statement:** This study did not require ethical approval.

**Informed Consent Statement:** Not applicable.

**Data Availability Statement:** Data can be asked from the corresponding author.

**Conflicts of Interest:** The authors declare no conflict of interest.

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
