# Peer review of "Two-Stage Fuzzy Traffic Congestion Detector"

_futuretransp, doi:10.3390/futuretransp3030047_

Round 1
Reviewer 1 Report
This paper presents a fuzzy-based method for short-term speed predictions in highways. The paper has clear research objectives and its methodology and results are sufficiently explained.
What is currently missing is a more detailed comparison with HCM and LOS method. In fact, authors already make a discussion on this between lines 409-436, but I believe that they should extend their discussion and better clarify the importance of their results by highlighting any policy implications and practical cases where their method should work better than the traditional ones.
Additionally, for Figures 2 they should also change the titles of their axes in English (they are in Italian now).
I also suggest to remove the "(then)" from their paper's title. It is not common to see words in parenthesis in papers' titles.
The quality of English is generally satisfactory. There are some editing and grammar mistakes which can be spotted and corrected relatively easily.
Author Response
Please see the attachment.
Best regards,
Gizem Erdinc

Reviewer 2 Report
In order to improve the quality of this study, some suggestions have been provided as below.
1.There are lots of prediction methods. Why did authors use this fuzzy one? Please highlight the motivation in introduction section.
2.From current situation, we cannot know the development of related works. Please add an additional literature review section.
3.The paper structure should be reorganized. Subsection 2.2 should be independent to a methodology section.
4.The title of sub-section 2.2.2 is "2.2.2. Subsubsection"? Please double check it.
5.From section 3, the results in Table 2 and Figure 8 seems good. But, this work need a comparison base. Please compare the results of fuzzy method to a non-fuzzy prediction model.
6.Discussion parts should be enhanced. So does conclusion section. In fact, authors didn't provide any meaningful discussion. Discussion section should be divided into a new section.
7.In conclusion, please include the implications of research and practices, limitations of used methods, and directions of future works.
8.Please update the cited works to the last published.
The quality of English should be edited by a native English speaker.
Author Response

(The authors gave the same response as above.)

Reviewer 3 Report
The paper proposes a two-step traffic state prediction using fuzzy logic based on real highway data, and proposes a calculation method for congestion degree, achieving short-term speed prediction and traffic state identification. However, this paper has some major issues and is not suitable for publication in this journal.
1. This paper utilizes fuzzy logic for speed prediction and traffic state identification. Many novel and more adaptive control algorithms have been applied to speed prediction and traffic state identification, such as machine learning. What are the advantages of fuzzy logic in predicting short-term speed and traffic status compared to more popular algorithms? In addition, as a well-known algorithm, fuzzy logic rules have been extensively introduced and explained in this paper.
2. The abstract mentions "a two-step fuzzy logic application...", but this paper only uses fuzzy logic in the first step of average speed prediction. The traffic state identification does not involve fuzzy logic, which leads to ambiguity in naming.
3. In line 112, it is mentioned that the traffic states are classified based on EU DATEX 113 standard ranges, but the concept and threshold of the traffic states are not unified as mentioned earlier. And line 44 mentions “there is still no unified approach …. to give our own approach “, which does not match before and after.
4. This paper uses historical traffic and density data to infer future road speeds. Using only historical 15 minutes road flow and density data when predicting the average speed in the next 15 minutes may not effectively characterize the trend of speed changes.
5. When displaying the results of road congestion: The speed prediction is based on a 15-minute interval, but Table 3 shows a total of 2.5 hours of congestion results from 6:30 to 9:00, which is too long and not practical.
6. The comparative methods are mentioned, but it is not well presented in the Application Results, which lacks support for the proposed method.
Additionally, there are many minor issues, such as:
1. Reference should avoid bulk citation.
2. Many variables in equations lack explanatory notes, such as Eq. (2). The interpretation of Flow in Eq. (1) is inconsistent. Is the Flow represented by "F" or "q", and is the density represented by "K" or "k"?
3. Regarding simulation, please provide details of the equipment used in the simulation and the timeliness of the simulation.
4. In Figure 2, the language used should be consistent with the language of the paper.
5. Some figures are blurry and unclear, eg. Figure 1, Figure 9.
Minor editing of English language required.
Author Response

(The authors gave the same response as above.)

Round 2
Reviewer 1 Report
The authors considered all my comments and made all relevant corrections and modifications in their paper. No further comments from my side.
Reviewer 2 Report
1.There are lots of prediction methods. Why did authors use this fuzzy one? Please highlight the motivation in introduction section. In revised version, authors didn't mention it and provide academic supports.
2.Authors claimed that "A comparison with the results of some relevant papers reported in the literature have been included and summarized in Table 4." But, only 3 tables in this manuscript.
3. In conclusion, please include the implications of research and practices.
4.Please update the cited works to the last published. From current version which uses tracking mode, no change could be found.
5.Authors are suggested to honestly respond all comments.
This study should be edited by a native English speaker.
Round 3
Reviewer 2 Report
All of my comments have been fully considered in this revised version.
The quality of English has been improved.